# The rise in ocean plastics evidenced from a 60-year time series

Clare Ostle [1], Richard C. Thompson[2], Derek Broughton[1], Lance Gregory [1], Marianne Wootton[1] & David G. Johns[1]

Plastic production has increased exponentially since its use became widespread in the 1950s. This has led to increased concern as plastics have become prevalent in the oceanic environment, and evidence of their impacts on marine organisms and human health has been highlighted. Despite their prevalence, very few long-term (>40 years) records of the distribution and temporal trends of plastics in the world's oceans exist. Here we present a new time series, from 1957 to 2016 and covering over 6.5 million nautical miles, based on records of when plastics have become entangled on a towed marine sampler. This consistent time series provides some of the earliest records of plastic entanglement, and is the first to confirm a significant increase in open ocean plastics in recent decades.

[1] The Marine Biological Association, The Laboratory, Citadel Hill, Plymouth PL1 2PB, UK. [2] School of Biological and Marine Sciences, University of Plymouth, Drake Circus, Plymouth PL4 8AA, UK. Correspondence and requests for materials should be addressed to C.O. (email: claost@mba.ac.uk)

Plastics have become a global concern as the accumulation in the world's oceans has become apparent, and potential health risks have been highlighted[1–4]. The use of man-made material for producing netting and fishing products became widespread in the 1950s, when these slow-degrading materials were made readily available and affordable[5,6]. Since this period, there has been a rapid increase in plastic production for a wide range of uses, and a number of environmental impacts have been highlighted[3,6–8]. Due to their very slow degradation periods, plastics have become ubiquitous and have been associated with marine health impacts[7] such as entanglement[9], ingestion[10,11], the potential dispersal of invasive species and toxicity[12], and contamination through trophic levels[13]. Comparable long-term environmental datasets of plastic debris are few and far between[4], with widespread data of sea surface plastic debris (particularly large plastic debris) being virtually non-existent[7]. Despite the increase in production of plastics and the consequent disposal, studies that have investigated open ocean plastic time series have been unable to show the expected increase since the 1990s[14–17].

Since 1957, the Continuous Plankton Recorder (CPR) has been towed over 6.5 million nautical miles in the North Atlantic and adjacent seas. The primary purpose of the CPR has been to record pelagic plankton, for which it is has been doing so since 1931 using ships-of-opportunity. Thompson et al.[14] used this historical record, to retrospectively count the amount of microplastics (plastic <5 mm)[3] within CPR samples along two transects in the northeast Atlantic. This study indicated a significant increase in microplastics from 1960–70 to 1980–1990, however no significant trend was observed between the 1980s and the 1990s[14]. The CPR consists of a torpedo shaped metal housing around a mechanical gear-shifted advancing mesh to collect and store the plankton, this design and technology has remained consistent since its inception in 1931[18]. The CPR is towed off the back of ships-of-opportunity, such as ferries and container ships, at ~7 m depth and from 10–20 knots speed[18]. In this sense, the CPR is susceptible to entanglement in a similar way to marine mammals that spend time in surface waters. When the CPR mechanism is hauled back on to the ship, the crew report any issues, such as entanglement on to a tow log, these are also noted and reported upon return to the CPR maintenance workshop.

Here we present a consistent time series from 1957 to 2016, of oceanic plastic occurrence in the North Atlantic and adjacent seas, and our findings on the localisation, properties, and temporal trends of these plastics. This dataset presents some of the earliest records of open ocean plastic entanglement, and confirms a significant increase in open ocean plastics since the 1990s[7,14–17].

## Results

**Observations**. Results presented here are based on observations that have been recorded as faults within a tow log for each CPR that has been towed within the North Atlantic and adjacent seas. 36% of the total number of CPR tows between 1957 and 2016 (16,725 tows) had faults logged, 4% of these faults were due to plastic entanglement and 1% were due to natural entanglement (see Methods and Supplementary information for further detail). Items were considered as macroplastic (plastic >5 mm)[3] if they were large enough to become entangled on the CPR, and described an assumed plastic item (see Supplementary Data 1 for each individual entanglement description).

**Macroplastics over the record**. Our analysis reveals an increase in macroplastic entanglement on the CPR since the first instance in 1957 (Fig. 1). In order to compare items that were not man-made a text search was also carried out for natural items, such as seaweed and wood that were reported to be entangled on the CPR (see Methods and Supplementary Data 2). Although there are gaps in both datasets, with more data being collected in the last 3 decades, the occurrence of natural items becoming entangled on the CPR remains consistent throughout the time series (Fig. 1a), and there was no significant correlation found between macroalgal entanglements and macroplastic entanglements (Student's $t$-test $p$-value < 0.05). The occurrence of macroplastic entanglement on the CPR shows a clear increasing trend with an order of magnitude increase from 2000 onwards (Fig. 1a, b). The records of macroplastics that became entangled on the CPR from 1957 to 2016 have a significant positive linear trend (standard model I linear regression) of $0.057 \pm 0.0066$ (Student's $t$-test $p$-value < 0.001 that trend is significantly different from zero), while there is no significant trend for the natural items (Fig. 1a). Figure 1b indicates a significant increase in macroplastics between the 1950s–60s, and the 1970–80 s, and both the 1990s and 2000s, respectively. Although we can not be sure on the trend until we have data for the whole decade in order to ensure comparability, 2010 to 2016 shows no significant change in total macroplastic counts compared to the 2000s.

**Regional trends and marine litter types**. The first record of man-made entanglement on the CPR was in 1957, recorded as recorder fouled by trawl twine, off the east coast of Iceland (Fig. 2). The second earliest record of man-made entanglement on the CPR was recorded as a plastic bag in 1965 off the Northwest of Ireland, this is the first mention of a specific plastic type becoming entangled on the CPR. The area that has the highest occurrence of macroplastic entanglement on the CPR is in the southern North Sea (Fig. 2, please see Supplementary Figs. 1–4 in the supplementary information for a regional break-down of trends). In the open ocean of the North Atlantic mostly netting, line and rope are reported (Fig. 2). The most commonly reported macroplastic within the CPR tow log is line (Supplementary Fig. 5). The term line is often reported as fishing line, with fishing net also being a main contributor to entanglement and becoming more prevalent in the tow logs since the early 1990s (Fig. 3, Supplementary Fig. 5 and Supplementary Data 1). Fishing gear was investigated by searching for the following terms; fishing, monofilament, trawl, and hook. Figure 3 indicates that the number of entanglements due to fishing related gear have increased, and that 55% of the total entanglements recorded are from fishing gear (see Supplementary Data 1). There is also evidence of a decline in the entanglement records of plastic bags since 2000 (Fig. 3). Applying the regions defined in Supplementary Fig. 1, a non-parametric (permutation-based) multivariate analysis of variance (perMANOVA) was carried out using the Fathom Matlab toolbox[19], which revealed a significant relationship ($p$-value < 0.05) between both year and region with litter category (see Methods for more detail). The majority of CPR tows were completed in the North Sea and Wider Atlantic area, giving greater statistical confidence in these areas (Supplementary Fig. 2). The increase in fishing related plastic entanglements, particularly in the North Sea region, contributed most significantly to the increase seen in macroplastic entanglements in the last 2 decades (Supplementary Fig. 4). Three of the five regions (Supplementary Fig. 1) presented a significant increasing trend in macroplastic entanglement (Student's $t$-test $p$-value < 0.001, Supplementary Fig. 3), with both the Bay of Biscay/Iberian Coast and the Arctic region giving a non-significant trend due to the lack of observations in these areas (Supplementary Fig. 2a, e). A similarity percentage (SIMPER) analysis[19] determined that the percentage contribution between the litter types to the change in macroplastic counts over time were 44.86% due to fishing related

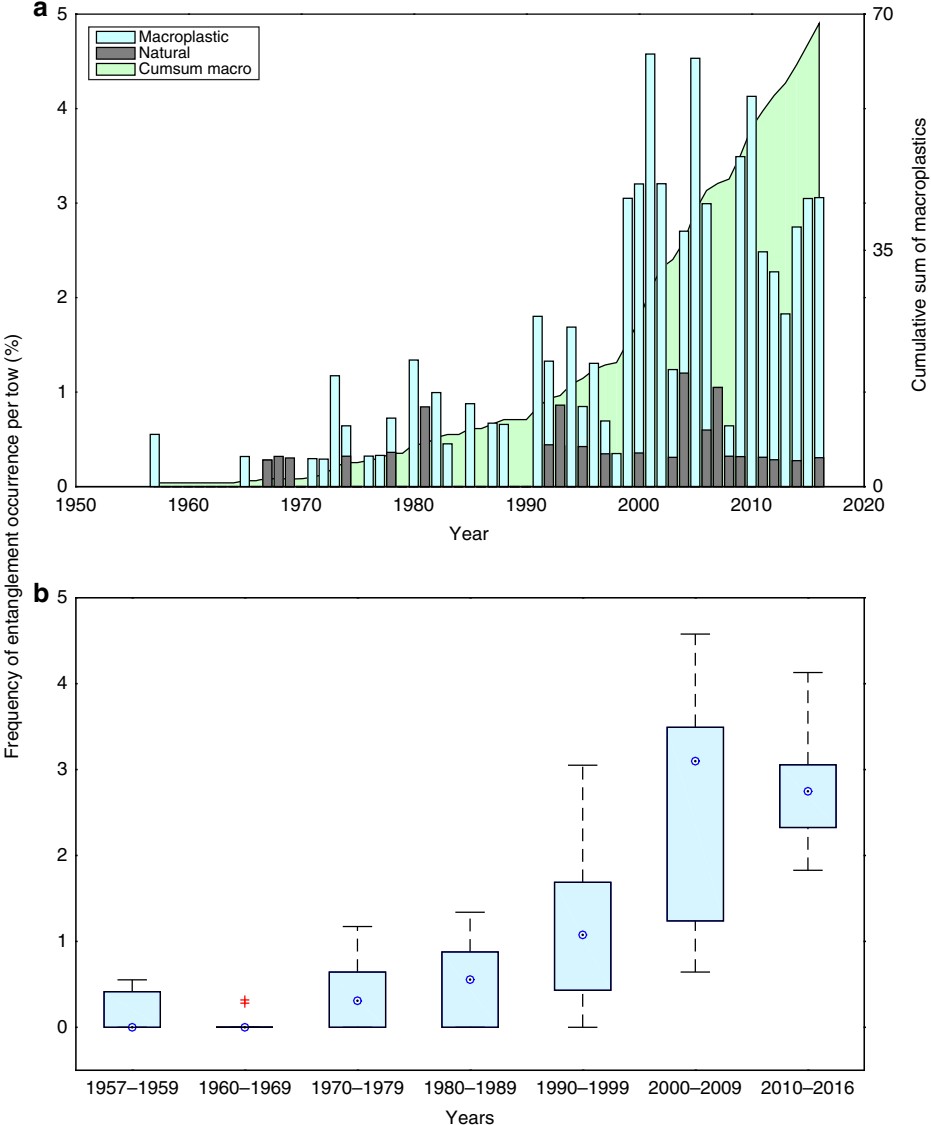

**Fig. 1** Total macroplastic and natural counts of entanglement. **a** annual counts of macroplastic = light blue bars (significant trend = 0.057 ± 0.0066 (1S.D.)), and natural items = grey bars (non-significant trend = 0.0045 ± 0.0028 (1S.D.)). The cumulative sum of the total counts of macroplastic are plotted on the second y-axis in green. **b** total counts of macroplastic per decade plotted as a box and whisker plot, where the median is represented by a blue circle, the edges of the box are the 25th and 75th percentiles, the whiskers extend to the most extreme data points, and outliers are plotted individually as red crosses. Please note there are only 3 years of data for the 1950s (1957–1959) and only 7 years of data for the 2010s (2010–2016). The annual counts of entanglements observed have been normalised to a percentage frequency of occurrence per Continuous Plankton Recorder tow (%)

plastics, 44.67% due to other (fishing not specified) plastic types, and 10.48% due to plastic bags.

## Discussion

Our findings are the first to confirm the expected significant increase in plastics in the open ocean since the 1990s[7,14–17]. We have presented a significant increase in macroplastics from 1957 to 2016 (Fig. 1), which agrees with the exponential increase in total plastic production worldwide[3]. It has been suggested that there may be a sink of plastic items within global oceans, which could have led to reduced estimates of sea surface plastics and have implications for plastic pollution[7,17]. Perhaps the reason we have been able to show the expected increase, is because the focus of this work has been on larger plastic items that entangle on the CPR. It should be noted that these larger plastics (macroplastics) break down under ultra-violet light and mechanical forces within

the ocean, leading to smaller fragments forming microplastics[20], therefore they have the potential to be a proxy for a wide-range of plastic sizes within the oceans.

There are very few historical records of oceanic plastic occurrence, in particular time series of macroplastics, with the most common reporting stemming from ingestion studies of seabirds and sea turtles[3,5,11,21–25]. In 1947 the earliest known reporting of entanglement was documented, when a herring gull was impeded by a piece of string[5,26]. However, it is uncertain if this string was made of plastic or natural fibres. In this study we have presented two early records of entanglement, one from 1957 when trawl twine was caught on the CPR and one in 1965 from plastic bag entanglement. The observations of trawl twine is synonymous with the widespread use of plastic for fishing practices in the 1950s as plastic became economical and more efficient to use than natural fibres[5,6]. Therefore although we cannot confirm that the twine was made from plastic, it is likely, as it was recorded in the

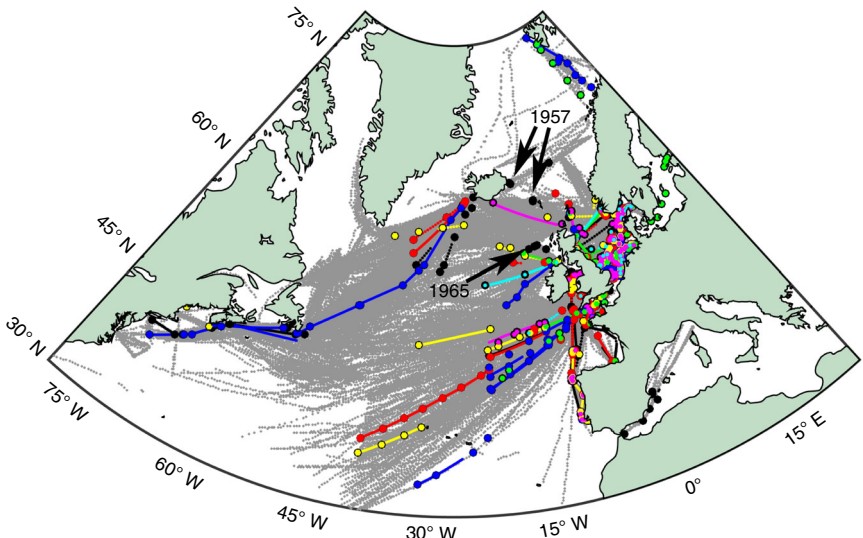

**Fig. 2** Map of macroplastics in the North Atlantic and adjacent seas. Map showing the location of Continuous Plankton Recorder samples since 1957 shown as grey points, black circles = macroplastic occurrence that has not been categorized, red circles = netting, blue circles = line, yellow circles = rope, green circles = bag, pink circles = monofilament, and cyan circles = string. The location of the two earliest records of macroplastic are annotated with the year they were recorded and black arrows. Note: The white area are where no data are available

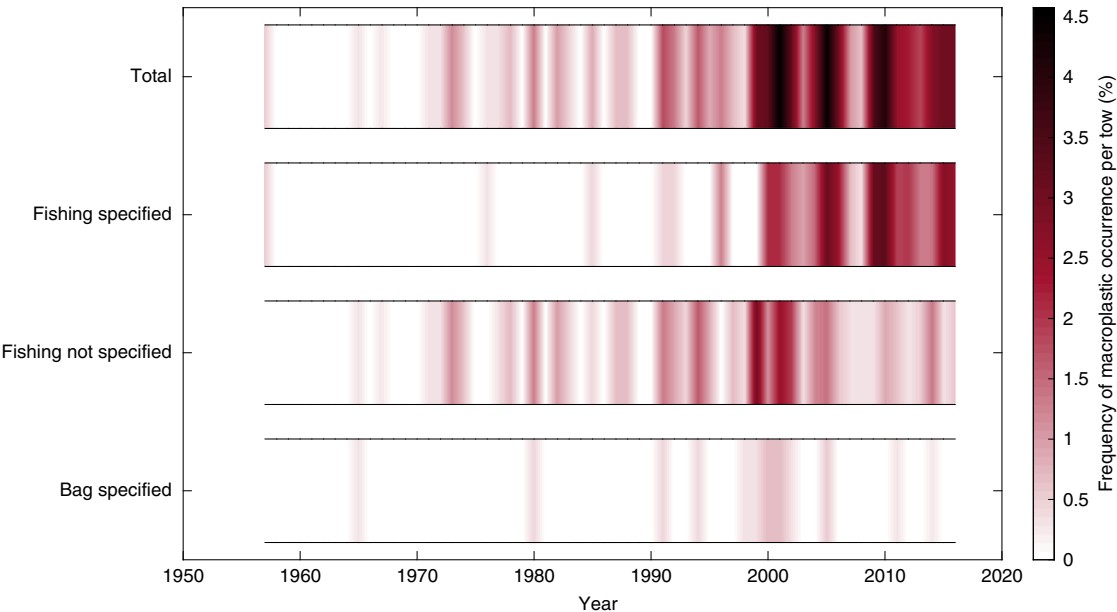

**Fig. 3** Annual macroplastic counts in categories from 1957 to 2016. Timeseries plot for four categories (total, fishing specified, fishing not specified, bag specified) of macroplastic entanglements from 1957 to 2016, with all macroplastics observed included in the total category. The annual counts of macroplastics observed have been normalised to a percentage frequency of occurrence per Continuous Plankton Recorder tow (%)

late 1950s and was specified as trawl twine. The record of a plastic bag entangled on the CPR in 1965 is within the same decade as the first known recordings of plastic entanglement and ingestion by seabirds[21–23], and marine turtles[7,24,25], which were observed in the late 1960s.

The data presented in Fig. 2 demonstrate that macroplastic debris are found throughout the North Atlantic. More macroplastic entanglements occurred in high-density shipping route areas, than areas of the open ocean such as the eastern North Atlantic, this could be due to the increased presence of human activity[6] introducing large plastic items to those areas (Fig. 2, and www.marinetraffic.com). Although the first record of man-made entanglement on the CPR in 1957 was recorded in Arctic waters

(Fig. 2 and Supplementary Fig. 1), we do not find a significant increase in macroplastic entanglements (Supplementary Fig. 3e), this is due to an increased number of CPR tows in the last decade in Arctic waters and our normalisation method employed to remove sampling bias (see methods for normalisation method and Supplementary Fig. 2e). However, we do record a peak in macroplastic entanglement cases in Arctic waters between 2009–2011 (Supplementary Fig. 3e), which corresponds with a significant increase in macroplastics between 2002–2014 seen at two stations at the HAUSGARTEN observatory in the Arctic[27]. Although not a significant increase, there were more plastic entanglement cases reported during January and December than any other month, suggesting that winter conditions such as high winds, rainfall and

river run-off, may have increased the amount of plastics within the oceans (Supplementary Fig. 4). This could partly explain the distribution of macroplastics seen in Fig. 2, where more entanglements have occurred near coastal and riverine input areas, such as the southern North Sea and the English Channel (Fig. 2 and Supplementary Fig. 4, perMANOVA $p$-value < 0.05[19]). These areas are also more likely to be impacted by anthropogenic pollution due to their close proximity to human populations[6].

Macroplastic items such as line and string are more likely to cause entanglement due to their shape (Fig. 3 and Supplementary Fig. 5). Our findings suggest that man-made entanglement from fishing related gear has significantly increased in recent decades (Fig. 3 and Supplementary Fig. 4), and could be more likely in areas such as the North Sea than the open ocean of the North Atlantic, where higher occurrences of macroplastics were reported to be entangled on the CPR (Fig. 2 and Supplementary Fig. 4, perMANOVA $p$-value < 0.05[19]). Although very few recreational fisheries are monitored in the North Sea, and commercial fish landings have reduced overall since the 1970s, data from the ICES fisheries overview indicates that pelagic trawl/seine fishing gear landings have increased since 2003 to 2015 in the North Sea[28]. The North Sea has large seabird and seal populations, as well as cetaceans, that are likely to have encountered plastic items and are at increasingly higher risk of entanglement[29,30].

As the global population continues to increase, plastic waste will continue to grow[31]. The realisation that plastics are ubiquitous, and that the consequent health impacts are yet to be fully understood, has increased the awareness surrounding plastics. Plastics are likely to be used as an indicator of marine health in environmental monitoring to drive policy (e.g. the Marine Strategy Framework Directive[32]). However, progression to reduce the inputs of plastic into the ocean is required[7,33,34]. There is a need for re-education, continued research and awareness campaigns, in order to drive action from the individual as well as large-scale decisions on waste-management and product design[4,33–35]. The use of ships-of-opportunity is an efficient (both in cost and time) way of covering vast areas, and should be utilised further to develop standardised and consistent strategies for monitoring plastic debris in the oceans[7]. The dataset presented here is an important historical record for the continued monitoring of plastics in the ocean, and confirms the importance of actions to reduce and improve plastic waste.

## Methods

**Data processing**. The observations presented within this manuscript are recorded within a digital database that dates back to 1957, they are descriptive comments with information recorded on the location, date and time that the CPR was towed. This information was gathered and categorized using a series of text queries to search for any words that are associated with man-made products within the tow logs, and to filter for those that are true cases of entanglement ($n = 208$). To extract observations of entanglement on the CPR via man-made items, the following words were used for a text search within the fault comments of the CPR database; plastic, net, line, rope, bag, fishing, twine, polyprop, monofilament, nylon, string. The observations that were extracted from the macroplastic search were checked for true cases of entanglement by manually reading through each extraction (see Supplementary Data 1 for all records). In order to compare these observations with items you might expect naturally to be found within the ocean that can entangle the CPR, the following words were used in a separate text search of the faults; seaweed, wood, fish, eel, bird (see Supplementary Data 2). For regional analyses the location of the end of the CPR tow was used to divide the dataset by region (see Supplementary Fig. 1), as this is more likely to be closer to the entanglement origination than the start of the CPR tow. The software package Matlab was used to process, carry out statistical analyses, and visualise the data presented. The investigation of linear trends within this study used standard model I linear regressions, and Student's $t$-test $p$-value < 0.001 to determine if the trend was significantly different from zero. Correlation analyses used the Pearson's correlation using a Student's $t$ distribution to determine the $p$-value. The Fathom toolbox[19] was used within Matlab to carry out a non-parametric (permutation-based) multivariate analysis of variance (perMANOVA) using a three-way Model III ANOVA with no replication, grouped by the litter type to investigate the trends between region (see Supplementary Fig. 1),

month, and year of each macroplastic entanglement (see Supplementary Fig. 4). The Fathom toolbox[19] was also used within Matlab to carry out a similarity percentage (SIMPER) analysis, to investigate the contribution of different litter types to the change in macroplastic entanglements.

**Normalisation**. A CPR tow is defined by when and where the ships' crew deploy (start of tow) and haul (end of tow) the CPR. Although the CPR is most commonly towed up to 480 nautical miles per tow, occasionally shorter tows are completed, which could introduce a sampling bias if the data were normalised to number of items per distance covered, as this change in sampling effort and potential increase in entanglement sighting may not be accounted for. Supplementary Fig. 6 demonstrates that the number of CPR tows is scalable to the distance towed by the CPR. In order to account for potential bias due to sampling effort, the counts of entanglements that have been recorded have been normalised for the number of CPR tows completed each year. To normalise the counts of macroplastic entanglement we used a percentage frequency of occurrence calculation = (number of entanglements/number of tows) × 100%. Supplementary Fig. 7 demonstrates the scalability of the macroplastic entanglements per area covered by the CPR (area ($m^2$) = distance towed (m) × width of CPR (0.225 m)), number of tows and distance towed, to normalise the observational data and provide comparative numbers.

## Data availability

Entanglement records are available in the Supplementary Data 1 (https://doi.org/10.17031/1617), and Supplementary Data 2 (https://doi.org/10.17031/1618). Annual macroplastic entanglements on the Continuous Plankton Recorder within each OSPAR region (note: The wider Atlantic region has been adapted from the OSPAR regions to extend to the west of the Atlantic to include the whole study area) are available in Supplementary Data 3 (https://doi.org/10.17031/08ga-a857), these are presented as per tow and per $m^2$ covered (note: The normalisation to area covered ($m^2$), may not take into account sampling bias of the number of tows completed). All data used within this publication can be accessed via www.cprsurvey.org, or by contacting the lead author via email at claost@mba.ac.uk.

## Code availability

The software package Matlab was used for data analysis and production of figures, these codes are available on reasonable request from the lead author via email at claost@mba.ac.uk.

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

## Acknowledgements
Funding that supports the running of the Volunteer Observing Shipping network used in this project includes DEFRA UK ME-5308, NSF USA OCE-1657887, DFO CA F5955-150026/001/HAL, NERC UK NC-R8/H12/100, IMR Norway, and DTU Aqua Denmark. We thank the people and funding agencies responsible for the maintenance and collection of data for their invaluable work and contributions.

## Author contributions
C.O. conceived the project. L.G., M.W. and D.G.J. maintain the operations and data collection. R.C.T. provided expertise on ocean plastics, and the data analysis. D.B. developed the dataset. C.O. carried out the computations and data analysis and prepared the first draft of the manuscript. All authors contributed to the discussion of the results and reviewed the manuscript.

## Additional information

**Competing interests:** The authors declare no competing interests.

