## [Peer Review File · Nature Communications]

Reviewers' comments:

Reviewer #1 (Remarks to the Author):

This manuscript is unique in the way that it provides important information about temporal trends in plastic pollution in the oceans, a currently very topical research field. It is potentially one of those land-mark papers and particularly important as (A) it comprises data over a very long time period spanning almost 50 years from 1957 to 2016, and (B) it manages to show an increase as we would expect to find given the tremendous increase in plastic production over the last 50 years. Other time series relying on repeatedly towed neuston nets have failed to show such a trend. In addition, the manuscript is very well written. I thus recommend publication but, I do have some concerns, which should be addressed in a revised version:

-My main concern is that the samples were taken over a very wide area (6.5 million nautical miles). One approach in time-series work is to repeatedly sample the same positions to be able to differentiate between spatial and temporal variability. On the other hand, some researchers think that covering a larger area makes estimates more robust. Although the authors address this in a way by adding Figure S4, this issue remains a little vague. I think advanced statistical testing with 'region' as a factor in addition to time would add power to the conclusions drawn.

To this end, in addition to Fig. 2, the authors should use multivariate statistics to test for differences in the litter composition over time, again with region as a sub-factor (e.g. via PERMANOVA, PRIMER-e). This would also allow them to tease out what litter category leads to differences between years (e.g. through the use of the SIMPER routine). This would add the deserved power to the now-slightly descriptive results. In Plymouth, the authors should have access PRIMER and experienced colleagues, who developed the programme.

-Another aspect worth of further testing: Is there a correlation between macroplastic and macroalgae? A lack of correlation would strengthen your argument that your results are not due to sampling bias.

-I like Figure 1, that is why it deserves to be described and discussed in a little more detail. For example, why do you think there is so little litter found in the southern part of the North Atlantic of the map? Why is there so little in the Greenlandic/Norwegian region but more in the Barents Sea/Arctic region? Why is there so much in the North Sea? Are there significant difference between regions (using time intervals as a factor)?

-The data are primarily presented as items per number of tows per year, which is fine for the context of this study. However, I strongly encourage the authors to provide a supplement with items per km as this would enable other workers to compare their data with this set. At a time, when everybody criticises the lack of standardisation in this research field, this really is important. In addition, Table

S1 should contain the positions (start/end) of each tow and a column giving the region according to S3. What does month mean? The survey month or the amount of months the CPR was at sea?

More specific comments:

-I think the reporting of the results should start with the first and second entanglement records and then with the highest records in the North Sea.

-This sentence belongs to Discussion: "This record is within the same decade as the first known recordings of plastic entanglement and ingestion by seabirds^{19–21}, and marine turtles^{7, 22, 23}, which were observed in the late 1960s."

-Another (albeit shorter) time-series study is one of the few to show a significant increase in macroplastic over time and could thus be added to the discussion (e.g. first paragraph), especially as it is from Arctic waters, where you also recorded litter in the same time period (2009-2011 peaks in Fig.S5e): Tekman MB, Krumpen T, Bergmann M. Marine litter on deep Arctic seafloor continues to increase and spreads to the North at the HAUSGARTEN observatory. *Deep-Sea Research I* 120, 88-99 (2017).

-There are quite a few errors/inconsistencies in the references cited (e.g. 4, 8, 13, 20), please go through these carefully and correct.

-All Figures given in nm, including Fig. S1, should be converted to common metrics, e.g. km or m rather than nautical miles

Reviewer #2 (Remarks to the Author):

Detecting a trend in ocean plastic debris over the decades since plastics entered widespread use (1950s) has been challenging, not least because of the relative dearth of environmental data. The authors present a creative analysis of opportunistic data on debris entanglements of a near-surface towed Continuous Plankton Recorder (CPR) device that have been collected in a consistent manner across a broad region of the North Atlantic Ocean since the late 1950s. Although the data set is somewhat limited by the lack of detailed description about each entanglement, and is unable to provide quantitative information about the increase in mass of ocean plastic debris, the analysis provides information about the increase in particular categories of oceanic debris that are likely to be composed of plastics (fishing line and nets, and bags). This information is valuable not only to potentially provide evidence of an increase in number of plastic debris items in time, but because it informs likely sources, which include both fishing activities and consumer use (e.g., bags).

I have reviewed a previous version of this paper, and the authors have addressed many of my comments. However, I still have questions about the specifics of the methodology, and the influence of potential sampling biases. Even if these biases cannot be quantified and resolved, they should be carefully considered and discussed.

Specifically, I don't understand why the number of CPR tows is equivalent to the distance towed (p. 1, last paragraph of Intro). And in Figure S1, it appears this is not actually the case since the two measures have different scales. How exactly is a tow defined, and why not simply use distance sampled as the relevant metric?

Do the crew report entanglement only when they bring the instrument aboard at the end of the voyage, or might they identify a fault, bring the instrument aboard and clear it, and then redeploy for a second deployment? In either case, how is the geographic position (lat/lon) of the entanglement determined? More information is needed about how tows are defined and entanglements recorded to ensure that comparisons across time periods are equivalent in (or normalized by) the distance sampled, which, I think, is a more relevant parameter than an arbitrarily defined tow.

If only 4% of faults were due to plastic entanglement and 1% were due to natural entanglement (p., 1, Results), does this mean the source of the remaining (95%) of faults could not be unambiguously determined?

With respect to the normalization by number of tows – the scales in the y-axes of Figures 3 and S3 seem way too large if the number of entanglements (211 reported across the entire record) is scaled by the number of tows (10^2 per year according to the axis in Fig S1). Similarly, I am having trouble reconciling the list of items in Table S1 and the data in Fig S3. For example, in ~1985 it seems there were a large number of macroplastic entanglements in the Arctic (25/tow), yet in the table there are only 2 entries for 1985. Perhaps there is a scale factor missing in these figures?

Further, I would like to know if there is a relationship between entanglement and ship speed. One could imagine that ships of opportunity have gotten faster over time, and that natural debris might break apart at higher ship speed (due to increased drag), whereas plastic line would not. Simple start/end dates and times for each tow (or each cruise) could be used to calculate approximate ship speed and determine if these have changed over time.

I think it should be clearly noted that the majority of tows, and thus the greatest statistical confidence, are in the North Sea and "Wider Atlantic" regions. Is the increase (in a region or overall)

correlated with increased fishing activity, especially from 2000 onwards (during largest observed increase)? The relationship with fishing activity could also be examined geographically – for example, it is very interesting that in the central portion of the Wider Atlantic region there were almost no entanglements due to macroplastic/fishing gear. Perhaps this is an artifact of incomplete records (i.e., source of fault not noted), but it would be interesting if this were related to a lower fishing effort in this region. If a relationship between fishing-related debris entanglements and fishing activity were evaluated, this could be a powerful conclusion to inform interventions or prevention efforts.

Response to reviewer's comments on Ostle *et al.*, The rise in Ocean Plastics: Evidence from a 60-year time series

We would like to thank the reviewers and editors involved for taking the time to review our manuscript and their thorough and constructive comments.

We have carefully considered all of the comments and changed the revised manuscript accordingly. In particular, we have addressed the potential sampling bias, and described the reasoning behind our normalisation process and regional differences more thoroughly. A non-parametric (permutation-based) multivariate analysis of variance (perMANOVA) analysis has been carried out to investigate the regional and litter-type changes over time, which we feel adds more weight to our findings and compliments the manuscript well.

In the following document we address each of the reviewer's comments, which are written in black text, with responses written in blue text. New references are included within the responses where appropriate.

Reviewer #1 (Remarks to the Author):

This manuscript is unique in the way that it provides important information about temporal trends in plastic pollution in the oceans, a currently very topical research field. It is potentially one of those land-mark papers and particularly important as (A) it comprises data over a very long time period spanning almost 50 years from 1957 to 2016, and (B) it manages to show an increase as we would expect to find given the tremendous increase in plastic production over the last 50 years. Other time series relying on repeatedly towed neuston nets have failed to show such a trend. In addition, the manuscript is very well written. I thus recommend publication but, I do have some concerns, which should be addressed in a revised version:

-My main concern is that the samples were taken over a very wide area (6.5 million nautical miles). One approach in time-series work is to repeatedly sample the same positions to be able to differentiate between spatial and temporal variability. On the other hand, some researchers think that covering a larger area makes estimates more robust. Although the authors address this in a way by adding Figure S4, this issue remains a little vague. I think advanced statistical testing with 'region' as a factor in addition to time would add power to the conclusions drawn. To this end, in addition to Fig. 2, the authors should use multivariate statistics to test for differences in the litter composition over time, again with region as a sub-factor (e.g. via PERMANOVA, PRIMER-e). This would also allow them to tease out what litter category leads to differences between years (e.g. through the use of the SIMPER routine). This would add the deserved power to the now-slightly

descriptive results. In Plymouth, the authors should have access PRIMER and experienced colleagues, who developed the programme.

Thank you for this valuable contribution. We have used the nonparametric (permutation-based) MANOVA software from the Fathom toolbox (Jones 2015), as we did not have direct access to PRIMER. The results indicate a significant relationship between both year and region with litter category, however no significant relationship with month sampled (month was added to investigate seasonal relationships). Following the SIMPER routines, the percentage contribution between the litter types were 44.86 % due to fishing related plastics, 44.67 % other plastic types, and 10.48% from plastic bags.

Following the regional analysis, 2 of the entanglements were reported outside of the defined regions shown in Fig. S4 (1 within the Mediterranean Sea, and 1 within the Baltic Sea), and 3 of the entanglements were found to be nearer the mid-Atlantic (not shown in Fig. 1). These 3 entanglement cases were excluded from the analysis for consistency with documenting primarily the North Atlantic.

The following text and figure (Fig. S7) have been added to the manuscript:

Line 34: *Results presented here are based on observations that have been recorded as faults within a tow log for each CPR that has been towed within the North Atlantic and adjacent seas.*

Line 39: *n=208*

Line 68: *Applying the regions defined in Figure S3, a non-parametric (permutation-based) multivariate analysis of variance (perMANOVA) was carried out using the Fathom Matlab toolbox (Jones2015), which revealed a significant ($p<0.05$) relationship between both year and region with litter category. The increase in fishing related plastic entanglements, particularly in the North Sea region, contributed most significantly to the increase seen in macroplastic entanglements in the last 2 decades (Fig. S7) The similarity percentage (SIMPER) analysis (Jones2015) determined that the percentage contribution between the litter types to the change in macroplastic counts over time were 44.86% due to fishing related plastics, 44.67% due to other (fishing not specified) plastic types, and 10.48% due to plastic bags.*

25. Jones, D. L. 2015. *Fathom Toolbox for Matlab: software for multivariate ecological and oceanographic data analysis*. College of Marine Science, University of South Florida, St. Petersburg, FL, USA. Available from: <http://www.marine.usf.edu/user/djones/>

-Another aspect worth of further testing: Is there a correlation between macroplastic and macroalgae? A lack of correlation would strengthen your argument that your results are not due to sampling bias.

Thank you for this suggestion. We have run a correlation analysis between the total macroplastic counts and the entanglement of macroalgae using both the f statistic, and t-test; neither gave a significant p-value.

The following text has been added to the manuscript:

Line 59: *Although there are gaps in both datasets, with more data being collected in the last 3 decades, the occurrence of natural items becoming entangled on the CPR remains consistent throughout the time series (Fig. 3a), and there was no significant correlation found between macroalgal entanglements and macroplastic entanglements (student's t-test pvalue <0.05).*

-I like Figure 1, that is why it deserves to be described and discussed in a little more detail. For example, why do you think there is so little litter found in the southern part of the North Atlantic of the map? Why is there so little in the Greenlandic/Norwegian region but more in the Barents Sea/Arctic region? Why is there so much in the North Sea? Are there significant difference between regions (using time intervals as a factor)?

Thank you. We have added more description to the discussion and the statistical significance between regions following the perMANOVA analysis from your previous suggestion. The following text has been added/updated:

Line 87: *The data presented in Figure 1 demonstrate that macroplastic debris are found throughout the North Atlantic. More macroplastic entanglements occurred in high-density shipping route areas, than areas of the open ocean such as the eastern North Atlantic, this could be due to the increased presence of human activity⁶ introducing large plastic items to those areas (Fig. 1, and www.marinetraffic.com).*

Line 89: *Although not a significant increase, there were more plastic entanglement cases reported during January and December than any other month, suggesting that winter conditions such as high winds, rainfall and river run-off, may have increased the amount of plastics within the oceans (Fig. S7). This could partly explain the distribution of macroplastics seen in Figure 1, where more entanglements have occurred near coastal and riverine input areas, such as the southern North Sea and the English Channel (Figs. 1 and S7, perMANOVA $p < 0.05^{25}$). These areas are also more likely to be impacted by anthropogenic pollution due to their close proximity to human populations⁶.*

Line 90: *Macroplastic items such as line and string are more likely to cause entanglement due to their shape (Fig. 2 and Fig. S1). Our findings suggest that man-made entanglement from fishing related gear has significantly increased in recent decades (Figs. 2 and S7), and could be more likely in areas such as the North Sea than the open ocean of the North Atlantic, where higher occurrences of macroplastics were reported to be entangled on the CPR (Figs. 1 and S7, per- MANOVA $p < 0.05^{25}$).*

-The data are primarily presented as items per number of tows per year, which is fine for the context of this study. However, I strongly encourage the authors to provide a supplement with items per km as this would enable other

workers to compare their data with this set. At a time, when everybody criticises the lack of standardisation in this research field, this really is important.

Thank you for this suggestion. We have added a supplementary figure (Fig. S3) that presents the percentage frequency of macroplastic entanglements per km. The reason we normalise the entanglements to the number of tows is because there could potentially be a sampling bias; if there were more short tows carried out, that would not be picked up by normalising by distance towed. Please see comments and additions made from reviewer 1.

-In addition, Table S1 should contain the positions (start/end) of each tow and a column giving the region according to S3. What does month mean? The survey month or the amount of months the CPR was at sea?

We have added both the start/end positions and a column to inform on region. The tables are now too wide to include within the supplementary information as tables, so they have been given a DOI (stored within the UK Archive for Marine Species and Habitats Data (DASSH)) and added as excel files. The month is the month that the tow was carried out, the headers of the table have been changed within the excel file to clarify this. We have also adjusted the location of the final reporting of the observation to the end of the tow, as this is the most likely location that the entanglement would have occurred. This has altered our regional analyses slightly (more samples reported within the Arctic, see Fig. S6e), however the main conclusions have not changed. The following wording in the manuscript has been adapted to reflect these changes:

Line 112: *For regional analyses the location of the end of the CPR tow was used to divide the dataset by region, as this is more likely to be closer to the entanglement origination than the start of the CPR tow.*

Line 114: *Entanglement records are available in the Supplementary Data 1 (<https://doi.org/10.17031/1617>), and Supplementary Data 2 (<https://doi.org/10.17031/1618>).*

More specific comments:

-I think the reporting of the results should start with the first and second entanglement records and then with the highest records in the North Sea.

Agreed, we have moved these sentences around, as suggested.

-This sentence belongs to Discussion: “This record is within the same decade as the first known recordings of plastic entanglement and ingestion by seabirds^{19–21}, and marine turtles^{7, 22, 23}, which were observed in the late 1960s.”

Agreed, we have moved this sentence to the discussion.

-Another (albeit shorter) time-series study is one of the few to show a

significant increase in macroplastic over time and could thus be added to the discussion (e.g. first paragraph), especially as it is from Arctic waters, where you also recorded litter in the same time period (2009-2011 peaks in Fig.S5e): Tekman MB, Krumpfen T, Bergmann M. Marine litter on deep Arctic seafloor continues to increase and spreads to the North at the HAUSGARTEN observatory. *Deep-Sea Research I* 120, 88-99 (2017).

Thank you for alerting us to this valuable study; we have added this to the discussion surrounding the regional analyses:

Line 88: Although the first record of man-made entanglement on the CPR in 1957 was recorded in Arctic waters (Fig. 1 and S4), we do not find a significant increase in macroplastic entanglements (Fig. S6e), this is due to an increased number of CPR tows in the last decade in Arctic waters and our normalisation method employed to remove sampling bias (Fig. S5e). However, we do record a peak in macroplastic entanglement cases in Arctic waters between 2009-2011 (Fig. S5e), which corresponds with a significant increase in macroplastics between 2002 - 2014 seen at two stations at the HAUSGARTEN observatory in the Arctic²⁴.

24. Tekman, M. B., Krumpfen, T. & Bergmann, M. Marine litter on deep Arctic seafloor continues to increase and spreads to the North at the HAUSGARTEN observatory. *Deep. Res. Part I Oceanogr. Res. Pap.* **120**, 88–99 (2017).

-There are quite a few errors/inconsistencies in the references cited (e.g. 4, 8, 13, 20), please go through these carefully and correct.

Apologies for this, we have gone through and corrected errors in the references.

-All Figures given in nm, including Fig. S1, should be converted to common metrics, e.g. km or m rather than nautical miles

Thank you for this suggestion, we have changed metric to km instead of nautical miles in figure S1 (now S2), please see further comments below as to why the data were normalised by tow instead of distance to avoid sampling bias.

Reviewer #2 (Remarks to the Author):

Detecting a trend in ocean plastic debris over the decades since plastics entered widespread use (1950s) has been challenging, not least because of the relative dearth of environmental data. The authors present a creative analysis of opportunistic data on debris entanglements of a near-surface towed Continuous Plankton Recorder (CPR) device that have been collected in a consistent manner across a broad region of the North Atlantic Ocean since the late 1950s. Although the data set is somewhat limited by the lack of detailed description about each entanglement, and is unable to provide quantitative information about the increase in mass of ocean plastic debris,

the analysis provides information about the increase in particular categories of oceanic debris that are likely to be composed of plastics (fishing line and nets, and bags). This information is valuable not only to potentially provide evidence of an increase in number of plastic debris items in time, but because it informs likely sources, which include both fishing activities and consumer use (e.g., bags).

I have reviewed a previous version of this paper, and the authors have addressed many of my comments. However, I still have questions about the specifics of the methodology, and the influence of potential sampling biases. Even if these biases cannot be quantified and resolved, they should be carefully considered and discussed.

- Specifically, I don't understand why the number of CPR tows is equivalent to the distance towed (p. 1, last paragraph of Intro). And in Figure S1, it appears this is not actually the case since the two measures have different scales. How exactly is a tow defined, and why not simply use distance sampled as the relevant metric?

A CPR tow is defined as when the ships' crew deploy the CPR and then haul the CPR back on to deck. Upon deployment and hauling of the CPR, the crew records the lat/lon and date/time for that tow. The CPR silk has a gear-shifted rotation that allows the CPR survey to determine the length of tow that has been completed. The number of CPR tows is broadly equivalent to the distance towed, this is due to the nature of the CPR methodology, whereby each plankton sample represents 10 nautical miles of tow, and each CPR cassette is towed for ~480 nautical miles. This should have been described more clearly in the text, therefore in addition to the figure in the supplementary material to demonstrate the relationship between number of tows and km towed (now Figure S2 with km instead of nautical miles), the following text has been added:

Line 55: A CPR tow is defined by when and where the ships' crew deploy (start of tow) and haul (end of tow) the CPR. Although the CPR is most commonly towed up to 480 nautical miles per tow, occasionally shorter tows are completed, which could introduce a sampling bias if the data were normalised to number of items per distance covered, as this increased effort may not be accounted for. In order to account for potential bias due to sampling effort, the counts of entanglements that have been recorded have been normalised for the number of CPR tows completed each year (see Fig. S2 for a comparison of number of CPR tows and distance towed in km).

Do the crew report entanglement only when they bring the instrument aboard at the end of the voyage, or might they identify a fault, bring the instrument aboard and clear it, and then redeploy for a second deployment? In either case, how is the geographic position (lat/lon) of the entanglement determined? More information is needed about how tows are defined and entanglements recorded to ensure that comparisons across time periods are

equivalent in (or normalized by) the distance sampled, which, I think, is a more relevant parameter than an arbitrarily defined tow.

Yes, the crew would report an entanglement when they haul the CPR, it is possible but unlikely that they would haul the CPR due to an entanglement. It is for this reason that we used number of tows, rather than towed distance/speed, to normalize the dataset and account for potential sampling bias. Because, if the tow was ended prematurely by a large entanglement, then a new tow would be initiated following the haul and clean-up. Also, if a number of shorter than normal (<480 nm) tows were carried out, then the crew would be more likely to report a higher number of entanglement cases (each time the CPR is hauled). Normalizing the dataset by towed distance or ship speed would not account for this potentially increased sampling effort. Please see previous comment for changes made in the manuscript to reflect this.

As suggested by reviewer 1, we have added a supplementary figure (Fig. S3) that presents the percentage frequency of macroplastic entanglements per km for comparison.

We have also adjusted the location of the final reporting of the observation to the end of the tow, as this is the most likely location that the entanglement would have occurred. This has altered our regional analyses slightly (more samples reported within the Arctic, see Fig. S6e), however the main conclusions have not changed. Both the location of the start of the tow and the end of the tow are now reported within the supplementary data files:

Line 112: *For regional analyses the location of the end of the CPR tow was used to divide the dataset by region, as this is more likely to be closer to the entanglement origination than the start of the CPR tow.*

Line 114: *Entanglement records are available in the Supplementary Data 1 (<https://doi.org/10.17031/1617>), and Supplementary Data 2 (<https://doi.org/10.17031/1618>).*

If only 4% of faults were due to plastic entanglement and 1% were due to natural entanglement (p., 1, Results), does this mean the source of the remaining (95%) of faults could not be unambiguously determined?

The majority (95%) of the faults recorded are not due to entanglement, but are due to issues with the CPR itself or attachments for the CPR, for example sometimes the CPR survey fit additional sensors to the CPR that may have had a fault recorded. These remaining faults were not reported in this manuscript as they are not relevant.

With respect to the normalization by number of tows – the scales in the y-axes of Figures 3 and S3 seem way too large if the number of entanglements (211 reported across the entire record) is scaled by the number of tows (10^2 per year according to the axis in Fig S1). Similarly, I am having trouble reconciling the list of items in Table S1 and the data in Fig S3. For example, in ~1985 it seems there were a large number of macroplastic entanglements in the Arctic

(25/tow), yet in the table there are only 2 entries for 1985. Perhaps there is a scale factor missing in these figures?

Apologies for this, we have not described the normalisation process very clearly in the manuscript, which explains the missing scale factor that you are describing. When normalising the counts of macroplastic faults to the number of tows, we used a percentage frequency of occurrence calculation:

$\% \text{ frequency of occurrence} = ((\text{number of entanglements} / \text{number of tows}) * 100\%)$

All relevant axes labels and the following text have been added to reflect this within the manuscript more clearly:

Para 55 final sentence: *To normalise the counts of macroplastic entanglement we used a percentage frequency of occurrence calculation = (number of entanglements/number of tows) x 100% (see Fig. S3 for macroplastic entanglements per tow and per kilometre).*

Further, I would like to know if there is a relationship between entanglement and ship speed. One could imagine that ships of opportunity have gotten faster over time, and that natural debris might break apart at higher ship speed (due to increased drag), whereas plastic line would not. Simple start/end dates and times for each tow (or each cruise) could be used to calculate approximate ship speed and determine if these have changed over time.

Thank you for this suggestion. We have followed your suggestion to investigate ship speed using the start and end times for each entanglement recorded. The long-term mean speed of all the ships was 28 km per hour. The earliest records in the 50s and 60s were at about 20 km per hour whereas many of the more recent ships recorded faster speeds, however many of the ships in recent decades also recorded similar slower speeds (this may be due to a push towards slow steaming for fuel economy). Whilst we do see a slight increase in ships' speed in recent decades, over the same time period, there is no apparent trend (decrease or increase) in natural entanglements, and no correlation with ship speed. This would suggest that the change in ships' speed is not significantly impacting natural entanglements.

I think it should be clearly noted that the majority of tows, and thus the greatest statistical confidence, are in the North Sea and "Wider Atlantic" regions.

Agreed, along with the text describing the additional perMANOVA analysis with region as a sub-factor (please see reviewer 1 comments), the following text has been added to the manuscript:

Within para 68: *The majority of CPR tows were completed in the North Sea and Wider Atlantic area, giving greater statistical confidence in these areas (Fig. S5).*

Is the increase (in a region or overall) correlated with increased fishing activity, especially from 2000 onwards (during largest observed increase)? The relationship with fishing activity could also be examined geographically – for example, it is very interesting that in the central portion of the Wider Atlantic region there were almost no entanglements due to macroplastic/fishing gear. Perhaps this is an artifact of incomplete records (i.e., source of fault not noted), but it would be interesting if this were related to a lower fishing effort in this region. If a relationship between fishing-related debris entanglements and fishing activity were evaluated, this could be a powerful conclusion to inform interventions or prevention efforts.

Thank you for this fruitful suggestion, as you had previously suggested we have altered figure 2 to show those items that specify fishing activity, and as you rightly point out the period from 2000 onwards shows a large increase in entanglement cases due to fishing related items. This corresponds to the increase seen in the Greater North Sea (Fig. S6 and the added figure S7 (please see reviewer 1 comments)). It is difficult to get hold of accurate fishing activity data in the North Atlantic. We have been trying to access AIS (Automatic Identification System) shipping data to work out where the fishing vessels are going, however there is a charge for this data. Vessel Monitoring System (VMS) data is also heavily restricted. The web portal www.marinetraffic.com does however let the user select the live fishing vessel positions for the current time, but to extract the density maps for the year there is a cost. The live tracker map on marinetraffic does correspond to the map presented in the manuscript, with most fishing vessels congregating around the coast and shelf edges. This would need much further interrogation and access to the historic fishing vessel data to confirm, which would be beyond the scope of this study, but we would hope to further investigate for a future study. However we have added some discussion on this topic as although fish landings in the Greater North Sea have reduced since the 1970s, there have been some changes in the fishing gear primarily used, with Pelagic trawl/seine fisheries showing an increase from 2003-2015 (ICES. Greater North Sea Ecoregion – Ecosystem overview. *ICES Advice 2016* 1–22 (2016). doi:10.17895/ices.pub.3116).

Following this comment and comments from reviewer 1 the following analysis and text has been added to the manuscript:

Line 68: Applying the regions defined in Figure S4, a non-parametric (permutation-based) multivariate analysis of variance (perMANOVA) was carried out using the Fathom Matlab toolbox (Jones2015), which revealed a significant ($p < 0.05$) relationship between both year and region with litter category. The increase in fishing related plastic entanglements, particularly in the North Sea region, contributed most significantly to the increase seen in macroplastic entanglements in the last 2 decades (Fig. S7) The similarity percentage (SIMPER) analysis (Jones2015) determined that the percentage contribution between the litter types to the change in macroplastic counts over time were 44.86% due to fishing related plastics, 44.67% due to other (fishing not specified) plastic types, and 10.48% due to plastic bags.

Line 87: *The data presented in Figure 1 demonstrate that macroplastic debris are found throughout the North Atlantic. More macroplastic entanglements occurred in high-density shipping route areas, than areas of the open ocean such as the eastern North Atlantic, this could be due to the increased presence of human activity⁶ introducing large plastic items to those areas (Fig. 1, and www.marinetraffic.com).*

Line 90: *Macroplastic items such as line and string are more likely to cause entanglement due to their shape (Fig. 2 and Fig. S1). Our findings suggest that man-made entanglement from fishing related gear has significantly increased in recent decades (Figs. 2 and S7), and could be more likely in areas such as the North Sea than the open ocean of the North Atlantic, where higher occurrences of macroplastics were reported to be entangled on the CPR (Figs. 1 and S7, per- MANOVA $p < 0.05^{25}$).*

Reviewers' comments:

Reviewer #1 (Remarks to the Author):

Dear authors,

thanks for taking on most of the points made but I am afraid that some issues remain to need addressing.

L.34-38 of 'Results' are actually a method description, which does not belong here. It should either go into methods or be integrated concisely into the introduction.

-Please start with the main finding of your study, which is that you have observed an increase in litter between 1957 and 2016

-Then please refer to Marine litter types and historical records (could actually be part of the time series record)

-I am missing a section on spatial differences. Why did you plot Figures S5/S6 if you do not describe them at all in the Results? Where are there differences between the areas in terms of litter quantity, trend, composition?

-Please describe in more detail how you did your statistics. Not the data extraction but the actual tests and software that you used.

Most importantly, please provide the data as plastic abundance per m or better still as per m².

Can you not estimate the area that your device covered? You do know the distance that the CPR has travelled and you do know the width of the device, so this should be possible. As mentioned in the last version, this is not a trivial request as this field of research currently lacks standardisation and the way the data are currently presented do not support comparison with other data, which is theoretically possible if graphs are modified so as to present data as per m². This way other workers can use the data to compare theirs to.

Reviewer #2 (Remarks to the Author):

The authors have addressed the concerns I raised in my review. I now better understand the choice to normalize by number of tows, although I think distance towed still needs to be considered because it could also lead to a sampling bias. For example, I think the authors mainly consider a potential sampling bias in terms of the number of opportunities (or likelihood) to detect a fault (e.g., the likelihood of detecting an entanglement across 480 nm is higher if it is sampled by 3 tows rather than 1). But one could also think of the bias in terms of the opportunity (or likelihood) of encountering debris (e.g., if one tow sampled only 120 nm compared to another tow that sampled 480 nm). In any case, looking at Figure S3 it seems the records are similar no matter which normalization one chooses.

I also appreciate the attempt to relate the increase in entanglements to a possible increase in fishing effort, and I agree that the work required is beyond the scope of the paper. Perhaps this would be a good starting point for a follow-on study.

Thanks to the authors for their attention to my concerns. I recommend the paper for publication.

Response to reviewer's comments on Ostle *et al.*, The rise in Ocean Plastics: Evidence from a 60-year time series

The authors would once again like to thank the editors involved and the reviewers for the time given to their thorough and constructive comments.

We feel that the addition of different units in figure S7, both in distance covered and area covered, adds valuable information for scalable comparisons with other marine litter and plastic studies.

In the following document we address each of the reviewer's comments, which are written in black text, with responses written in blue text. The changes that have been made within the manuscript and supplementary information are highlighted in yellow.

Reviewers' comments with responses:

Reviewer #1 (Remarks to the Author):

Dear authors,
thanks for taking on most of the points made but I am afraid that some issues remain to need addressing.

L.34-38 of Results are actually a method description, which does not belong here. It should either go into methods or be integrated concisely into the introduction.

We agree and have moved the paragraph into the methods section under '*Data processing*', line 119.

-Please start with the main finding of your study, which is that you have observed an increase in litter between 1957 and 2016. Then please refer to Marine litter types and historical records (could actually be part of the time series record)

Thank you for your suggestion, we think your suggested re-arrangement works well, and have now removed the '*historical records*' section, and added the '*regional trends*' into the '*marine litter types*' section, please see the following comment.

-I am missing a section on spatial differences. Why did you plot Figures S5/S6 if you do not describe them at all in the Results? Where there differences between the areas in terms of litter quantity, trend, composition?

Agreed, we have added the combined section '*Regional trends and marine litter types*', and included the following text to describe the spatial differences further. Note: The supplementary figure numbers have changed since the

previous manuscript version due to the re-order of the results section, figures S5 and S6 are now S2 and S3.

Line 66: *Applying the regions defined in Figure S1, a non-parametric (permutation-based) multivariate analysis of variance (perMANOVA) was carried out using the Fathom Matlab toolbox²⁵, which revealed a significant relationship ($p < 0.05$) between both year and region with litter category (see methods for more detail). The majority of CPR tows were completed in the North Sea and Wider Atlantic area, giving greater statistical confidence in these areas (Fig. S2). The increase in fishing related plastic entanglements, particularly in the North Sea region, contributed most significantly to the increase seen in macroplastic entanglements in the last 2 decades (Fig. S4). Three of the five regions (Fig. S1) presented a significant increasing trend in macroplastic entanglement (student's t-test pvalue < 0.001 , Fig. S3), with both the Bay of Biscay/Iberian Coast and the Arctic region giving a non-significant trend due to the lack of observations in these areas (Fig. S2a,e). A similarity percentage (SIMPER) analysis²⁵ determined that the percentage contribution between the litter types to the change in macroplastic counts over time were 44.86% due to fishing related plastics, 44.67% due to other (fishing not specified) plastic types, and 10.48% due to plastic bags.*

Spatial differences are also referred to in the discussion.

-Please describe in more detail how you did your statistics. Not the data extraction but the actual tests and software that you used.

We have added the following description of the tests and software used to the methods section 'Data processing':

Line 129: *The software package Matlab was used to process, carry out statistical analyses, and visualise the data presented. The investigation of linear trends within this study used standard model I linear regressions, and student's t-test p-value < 0.001 to determine if the trend was significantly different from zero. Correlation analyses used the Pearson's correlation using a Student's t distribution to determine the p-value. The Fathom toolbox²⁵ was used within Matlab to carry out a non-parametric (permutation-based) multivariate analysis of variance (perMANOVA) using a three-way Model III ANOVA with no replication, grouped by the litter type to investigate the trends between region (see Fig. S1), month, and year of each macroplastic entanglement (see Fig. S4). The Fathom toolbox²⁵ was also used within Matlab to carry out a similarity percentage (SIMPER) analysis, to investigate the contribution of different litter types to the change in macroplastic entanglements.*

Most importantly, please provide the data as plastic abundance per m or better still as per m². Can you not estimate the area that your device covered? You do know the distance that the CPR has travelled and you do know the width off the device, so this should be possible. As mentioned in the

last version, this is not a trivial request as this field of research currently lacks standardisation and the way the data are currently presented do not support comparison with other data, which is theoretically possible if graphs are modified so as to present data as per m². This way other workers can use the data to compare theirs to.

Thank you for this suggestion, the authors very much agree that having comparable standardised data is important. We have therefore followed your suggestion and calculated the area covered by the CPR, to give a number of plastic entanglements per m². This data is presented alongside the number of entanglements normalised per number of tows, and the number of entanglements per km covered in figure S7. As you would expect these numbers are small due to the relatively large number of tows and distance covered by the CPR, however we feel the data is still of importance and useful to present in this way. Please see the following text that we have added to and moved to the methods normalisation section, where we have described the method for calculating the area covered by the CPR:

Line 130: A CPR tow is defined by when and where the ships' crew deploy (start of tow) and haul (end of tow) the CPR. Although the CPR is most commonly towed up to 480 nautical miles per tow, occasionally shorter tows are completed, which could introduce a sampling bias if the data were normalised to number of items per distance covered, as this change in sampling effort and potential increase in entanglement sighting may not be accounted for. Figure S6 demonstrates that the number of CPR tows is scalable to the distance towed by the CPR. In order to account for potential bias due to sampling effort, the counts of entanglements that have been recorded have been normalised for the number of CPR tows completed each year. To normalise the counts of macroplastic entanglement we used a percentage frequency of occurrence calculation = (number of entanglements/number of tows) x 100%. Figure S7 demonstrates the scalability of the macroplastic entanglements per area covered by the CPR (area (m²) = distance towed (m) x width of CPR (0.225 m)), number of tows and distance towed, to normalise the observational data and provide comparative numbers.

Due to the way in which entanglements are reported on the CPR, the authors feel that the normalisation of the data is more accurate if normalised per number of tows rather than the distance covered, as an increased number of tows is more likely to increase the number of sightings of entanglements on the CPR and this potential sampling bias needs to be accounted for. Because, if the tow was ended prematurely by a large entanglement, then a new tow would be initiated following the haul and clean-up. Also, if a number of shorter than normal (<480 nm) tows were carried out, then the crew would be more likely to report a higher number of entanglement cases (each time the CPR is hauled). Normalising the dataset by towed distance/area or ship speed would not account for this potentially increased sampling effort. However, when comparing the macroplastic entanglements that have been normalised by

distance towed with the normalisation using number of tows (please see figures S6 and S7 in the supplementary information), there is not a significant difference in the trends, and it is scalable due to the CPR methodology.

Reviewer #2 (Remarks to the Author):

The authors have addressed the concerns I raised in my review. I now better understand the choice to normalize by number of tows, although I think distance towed still needs to be considered because it could also lead to a sampling bias. For example, I think the authors mainly consider a potential sampling bias in terms of the number of opportunities (or likelihood) to detect a fault (e.g., the likelihood of detecting an entanglement across 480 nm is higher if it is sampled by 3 tows rather than 1). But one could also think of the bias in terms of the opportunity (or likelihood) of encountering debris (e.g., if one tow sampled only 120 nm compared to another tow that sampled 480 nm). In any case, looking at Figure S3 it seems the records are similar no matter which normalization one chooses.

I also appreciate the attempt to relate the increase in entanglements to a possible increase in fishing effort, and I agree that the work required is beyond the scope of the paper. Perhaps this would be a good starting point for a follow-on study.

Thanks to the authors for their attention to my concerns. I recommend the paper for publication.

Thank you very much for your contributions.

REVIEWERS' COMMENTS:

Reviewer #1 (Remarks to the Author):

Dear authors,
thank you for the latest revision , which I am now happy to strongly recommend for publication in Nature Comm!

One recommendation I still have: Please, supply a Table with data per region giving the actual values as per m² for the regions. This could be either in the supplement or in the form of archived data, which is open access. This is now common data policy of many institutions. It allows accessibility of comparable data and enables other researchers to compare accurate data rather than having to read values of graphs.

REVIEWERS' COMMENTS:

Reviewer #1 (Remarks to the Author):

Dear authors,
thank you for the latest revision , which I am now happy to strongly recommend for publication in Nature Comm!

One recommendation I still have: Please, supply a Table with data per region giving the actual values as per m² for the regions. This could be either in the supplement or in the form of archived data, which is open access. This is now common data policy of many institutions. It allows accessibility of comparable data and enables other researchers to compare accurate data rather than having to read values of graphs.

Thank you for this positive suggestion, we have now added a 3rd supplementary data table as an excel file, including the data per region as per m² as suggested. The following text has been added to the data availability statement within the manuscript to reflect this:

Annual macroplastic entanglements on the Continuous Plankton Recorder within each OSPAR region (note: The wider Atlantic region has been adapted from the OSPAR regions to extend to the west of the Atlantic to include the whole study area) are available in Supplementary Data 3 (<https://doi.org/10.17031/08ga-a857>), these are presented as per tow and per m² covered (note: The normalisation to area covered (m²), may not take into account sampling bias of the number of tows completed).